# Detection and attribution of quotes in Finnish news media: BERT vs. rule-based approach

**Maciej Janicki** and **Antti Kanner** and **Eetu Mäkelä**
Department of Digital Humanities
University of Helsinki
Unioninkatu 40, 00170 Helsinki, Finland
`firstname.lastname@helsinki.fi`

## Abstract

We approach the problem of recognition and attribution of quotes in Finnish news media. Solving this task would create possibilities for large-scale analysis of media wrt. the presence and styles of presentation of different voices and opinions. We describe the annotation of a corpus of media texts, numbering around 1500 articles, with quote attribution and coreference information. Further, we compare two methods for automatic quote recognition: a rule-based one operating on dependency trees and a machine learning one built on top of the BERT language model. We conclude that BERT provides more promising results even with little training data, achieving 95% F-score on direct quote recognition and 84% for indirect quotes. Finally, we discuss open problems and further associated tasks, especially the necessity of resolving speaker mentions to entity references.

## 1 Introduction

The recognition of quotes and reported speech is an important step towards the computational analysis of news media articles. It allows us to measure, on a large scale, who is given voice and how much, how opposing or competing views are presented alongside each other, as well as how the language of the quoted sources differs from the language of the journalistic reporting. In case of the Finnish news media, such analyses have recently been attempted by (Koivunen et al., 2021; Seuri et al., 2021). On the other hand, Suomen Kuvalehti et al. (2021) have studied politicians' visibility in the media based on the mentions of their names.

In the present paper, we focus on the technical task of recognizing direct and indirect quotes in the Finnish news media texts. The task can be illustrated with the following example:

> **Sipilän** mukaan *lakiehdotuksia ollaan tuomassa eduskuntaan helmikuussa.*

> According to **Sipilä**, *bill proposals will be brought to the parliament in February.*

Such relations consists of three elements: the **cue** 'mukaan' ('according to') indicates an indirect quote, in which the **source** (Juha Sipilä, the Finnish prime minister 2015–2019) says the text referred to as **proposition**, or **quotation span**.[1] A complete approach for quote detection and attribution would solve the following tasks:

1. Detecting quotation spans.

2. Attributing quotation spans to the source mention in the text (which might also span multiple tokens).

3. Linking source mentions to entity identifiers (including coreference resolution and lemmatization).

We will present methods for solving tasks 1 and 2, while discussing 3 as subject for further work.

Most existing work for this task deals with English, while occasionally other Germanic or Romance languages have been considered. Compared to that, Finnish presents challenges due to a rich morphology and free word order. Those can largely be dealt with by the advanced NLP tools that we are using (either a dependency parser pipeline or BERT), but they rule out the usage of simpler pattern-based methods and remain a possible source of errors even for state-of-the-art NLP.

---

[1]We follow Pareti (2015)'s convention of marking the quotation span in cursive, the source in bold, and underlining the cue.

We describe the process of collecting and annotating a gold standard corpus in Sec. 3. Further, in Sec. 4, we describe two different automatic approaches: a rule-based one, amounting to matching certain grammatical structures in dependency-parsed text, as well as a machine learning one, which utilizes the state-of-the-art neural language model BERT. The corpus and the code for both methods are publicly available.[2] [3] [4]

Our initial intuition was that dependency parsing provides enough information to recognize quotes with simple pattern matching. Another reason to implement this approach was that it did not need training data, which was at first unavailable for us. However, the final comparison revealed that the BERT-based model outperformed the rule-based even with little training data. The results of this experiment are described in Sec. 5.

## 2 Related Work

To our knowledge, the most similar work to ours has been done by Silvia Pareti and colleagues (Pareti et al., 2013; Pareti, 2015, 2016), who annotated a corpus of attribution relations for English and experimented with machine learning models for recognizing such relations. For the latter they applied classification algorithms – CRF, k-NN, logistic regression – working on data enriched with linguistic features, which was state-of-the art in NLP at the time. However, Scheible et al. (2016) have criticized the choice of CRFs for quote detection because of the Markov assumption they make. More recently, Papay and Padó (2019) presented a neural LSTM-based model for recognizing quotations, but without attribution. Brunner et al. (2020) compare different embedding-based models (including BERT) on the task of recognizing types of speech, which includes direct and indirect quotes.

As to Nordic languages, a rule-based approach for Norwegian has been presented by Salway et al. (2017). It utilizes a dependency parser and a list of speech verbs. From among other languages, Quintão (2014) used a machine learning method on Portuguese news corpora, while Pouliquen et al. (2007) used a rule-based approach for multiple European languages.

[2] https://github.com/hsci-r/
fi-quote-coref-corpus
[3] https://github.com/hsci-r/
flopo-quote-detection
[4] https://github.com/hsci-r/
flopo-quotes-bert

Muzny et al. (2017) present a method for quote attribution. They thus start with quotation spans already recognized and perform two tasks: 1) attributing a quote to a speaker mention in the text, 2) linking the speaker mentions into entities. They use a rule-based strategy on top of tools performing dependency parsing and coreference resolution. They have also released a corpus of quote attributions consisting of three novels in English.

Although not dealing exactly with quote detection, Padó et al. (2019) provide a prominent example of computational analysis of political discourse using modern NLP methods. They use various neural models (including BERT) to detect claims and attribute them to actors, with the goal of modeling the discourse as a network of relations between actors and claims. Automatic quote detection could be a useful element of such a larger system as well.

## 3 Dataset and Annotation

The annotation process consisted of two parallel tasks: marking quotations and linking together chains of co-referencing expressions denoting people, institutions and other human-like actors present in the documents. Both annotation tasks were conducted using the WebAnno platform (Eckart de Castilho et al., 2016), by which each annotator was assigned their documents and by which the annotation itself was done. The annotation guidelines were written beforehand and further developed after a test run.

The quotation detection annotation consisted of 1) marking the span in the text containing the content of the quote, 2) marking the speech act verb (if present), 3) marking the source of the quotation (if present), and 4) noting whether the quote was direct or indirect. The task was relatively straightforward, as all annotators were students with at least a minor degree in linguistics.

The project employed 10 annotators. Four of them were recruited in an earlier phase and annotated a test data set of 40 articles. After the test run, the guidelines were improved based on both inter-annotator agreement scores and feedback from the annotators, in accordance with the standard linguistic annotation methodology (Artstein, 2017). The inter-annotator agreement scores (Fleiss' $\kappa$) were between 0.77-0.8, which we deemed sufficient to consider the annotations consistent. The workload was balanced so that the 6

other annotators who were recruited at the later stage annotated more articles to compensate for the test run. The annotators worked independently on the WebAnno platform.

The articles were sampled from a database containing the metadata for the online media sources and the sampled lists of articles were then scraped using a web crawler (Mäkelä and Toivanen, 2021) and automatically pre-processed to CONLL format containing lemmatization, part-of-speech and dependency taggings using Turku Neural Parser (Kanerva et al., 2018). We used four sources for the articles: YLE (the Finnish national broadcasting company), Helsingin Sanomat (the most popular daily newspaper), Iltalehti (an evening tabloid) and STT (the Finnish news agency), covering different kinds of media texts wrt. length and style. The total number of articles annotated was 1500. Except for the common part mentioned above, the remaining 1460 articles were assigned to one annotator each at the second stage.

## 4 Methods

### 4.1 Rule-based approach

The input to the rule-based quote detection engine is text with linguistic annotations obtained from the Turku Neural Parser (Kanerva et al., 2018).[5] The parser performs the following tasks: tokenization, lemmatization, part-of-speech and morphological tagging, and dependency parsing.

The first stage of quote recognition is recognizing syntactic structures that typically introduce a quote (Table 1). Rules 1-2 describe the very common structures like 'X says that Y' and 'Y, says X', respectively. Rules 3-4 describe structures of the type: 'according to X, Y' and 'in X's opinion, Y'. In such structures, the source and cue can be positioned differently relatively to the proposition: before, after, or even inside it (see the example for rule 4). In the latter case, we allow annotating the cue and source as part of the proposition to avoid discontinuous propositions. Finally, rule 5 is characteristic for Finnish: it captures the construction '*says* + active participle', e.g. *sanoo olevansa*

---

[5]A reviewer has plausibly remarked that using the dependency parser available in spaCy could simplify the architecture. We have not evaluated the impact of this change on performance, as at the time of implementing the method Turku Neural Parser was considered state-of-the-art for Finnish and, unlike spaCy, the Turku parser was applied in various other ways in the project context. However, the rules are coded in the spaCy DependencyMatcher format, so they can easily be tried on spaCy output as well.

'says that he is', or *sanoo tehneensä* 'says that he did'. This construction does not use the word *että* 'that'.

In the rules where the cue is a verb (1, 2 and 5), the verb *sanoa* 'to say' can be substituted by any other speech act verb, e.g. *kertoa* 'to tell', *korostaa* 'to emphasize', *kuitata* 'to sum up' etc. We initially prepared a list of speech act verbs manually, then used a word2vec model to expand it with automatically generated synonyms, which were again filtered manually. The final list consisted of 73 verbs.

Once the source-cue-proposition triplets are recognized, the proposition texts can typically be extracted by taking the dependency subtree under the token marked as proposition. However, further post-processing is needed for quotes consisting of multiple sentences. For example in Table 1, the example for rule 2 is clearly the last sentence of a multi-sentence quote. In order to expand the matches to multi-sentence quotes, we use two rules:

1. If the paragraph containing the match starts with a hyphen – extend the quote to the beginning of the paragraph. This is because long direct quotes are typically formatted as separate paragraphs.

2. If there is a quotation mark between the cue and the proposition head – extend the quote backwards to the matching quotation mark.

In both these cases, the quote is classified as direct, as it is marked with quotation markers. Matches that do not fulfill the above conditions are classified as indirect.

Finally, we use an additional rule to detect 'freestanding' direct quotes encompassing entire paragraphs. These do not necessarily contain a source attribution (like ', says X') because the source might be already clear from the context. Thus, we detect remaining paragraphs that either start with a hyphen or are enclosed in quotation marks, as direct quotes. For the attribution we currently use a naïve strategy of attributing them to the same source as the previous quote in the text (if present). This works in a lot of cases because the quotes usually follow a structure in which a whole-paragraph direct quote is introduced by a preceding sentence containing an indirect quote, like in the following example:

According to Lindberg, approximately every third pet is overweight.

– We do have a lot of work on that.

The rules from Table 1 are implemented using the spaCy library class `DependencyMatcher`[6] which offers a declarative language to express the rules and good performance. The post-processing code is implemented in Python.

### 4.2 BERT model

The machine learning model is realized as two token classification heads on top of BERT – a neural language model based on the transformer architecture (Devlin et al., 2019). We use the model pretrained on Finnish data by Virtanen et al. (2019).

The first classification head recognizes and classifies spans of quoted text (propositions). The labeling follows the IOB schema and the class label encodes whether the quote is direct or indirect, as well as the relative position of the speaker mention to the quoted text. The latter is expressed as one of the symbols: +, − or = and a number 1-4. The symbol describes whether the speaker is mentioned after (+), before (−) or inside (=) the proposition, while the number signifies, which recognized entity is the speaker. For example, the class label `B-DIRECT+2` denotes the *beginning* (B−) of a *direct* quote, the source of which is the *second* recognized entity *after* the quote. A special label `00` signifies that the source of the quote is not marked.

The second classification head recognizes the entities, i.e. elements of coreference chains. It has just one class encoded in the IOB schema and does not perform the linking of entities into chains.

An example of sequence annotation is shown in Table 2. It shows the following sentence:

> *Kansainvälinen rikostuomioistuin aikoo määrätä Sudanin presidentin Omar al-Bashirin pidatettäväksi,* kertoo **sanomalehti New York Times.**

> *The International Criminal Court is intending to issue an arrest warrant on Sudan's president Omar al-Bashar,* **the newspaper New York Times** reports.

There are three entities in the sentence: 'The International Criminal Court', 'Sudan's president

---

[6] `https://spacy.io/api/dependencymatcher`

Omar al-Bashar' and 'the newspaper New York Times' – their annotations on the token level are encoded on the 'entity' layer. The 'quote' layer encodes an indirect quote, which is attributed to the first entity following the quote (hence, +1).

## 5 Evaluation

For the evaluation experiments we use a roughly 80-20 split of the data by taking the data provided by 2 annotators as evaluation set and the remaining 8 annotators as training set. The dataset sizes are summarized in Table 3. We compare both methods on the task of quote recognition (with and without direct/indirect classification) and attribution.

**Quote detection.** The results of quote span detection without taking into account the direct-indirect distinction are shown in Table 4. On the other hand, the direct-indirect breakdown is shown in Table 5, where misclassifications (identifying a direct quote as an indirect one or vice versa) were counted as both a false positive and a false negative. We exclude punctuation tokens from the evaluation as especially the commas and periods on the boundaries of quotes might have been inconsistently annotated, and their inclusion in the quote is irrelevant.

Both settings show a clear advantage of the BERT model. In case of direct quotes, the rules for recognizing them are quite rigid. Furthermore, they can suffer from paragraph segmentation errors and misplaced or incidental quotation marks (e.g. 'scare quotes'). This explains the lower recall of the rule-based method.

Indirect quotes have proven more challenging to the rule-based method as well. This can be to a variety of reasons: missing speech act verbs, incorrectly identifying quote spans based on syntactic criteria (also affected by parser, tagger and sentence segmentation errors), or uncommon structures not covered by the rules. Moreover, rule 3 ('according to') has a tendency to produce false positives, e.g. something being described 'according to the plan'.

In general, the BERT model has shown to be more flexible wrt. the often unpredictable nature of text data, and does not suffer from the error propagation through the NLP pipeline.

**Attribution.** The evaluation of attribution is problematic because of the fact that our dataset was not annotated with the BERT model in mind.

| No. | schema | example |
|---|---|---|
| 1 | nsubj, ccomp
source cue prop
VERB | **Malinen** sanoo, että *hän ei tule esittämään liiton hallituk-selle yhdenkään sopimuksen hyväksymistä.*
**Malinen** says that *he will not propose accepting even a single motion of agreement to the union's board.* |
| 2 | nsubj, parataxis
source cue prop
VERB | *Siksi mekin lähdimme näihin neuvotteluihin mukaan,* **Mäkynen** sanoo.
*This is why we also joined these negotiations,* **Mäkynen** says. |
| 3 | obl, case
source cue prop
LEMMA: 'mukaan' | **Sipilän** mukaan *lakiehdotuksia ollaan tuomassa eduskun-taan helmikuussa.*
According to **Sipilä**, *bill proposals will be brought to the parliament in February.* |
| 4 | nmod:poss, (any)
source cue prop
LEMMA: 'mieli'
CASE: Ela | *Suomen vaikeista ongelmista talous on* **presidentin** mielestä *helpompi.*
*From Finland's most difficult problems, the economy is* in **the president's** opinion *easy.* |
| 5 | nsubj, xcomp
source cue prop
VERB | **Orpo** sanoo *olevansa valmis poikkeuksellisiin keinoihin ja jopa lainmuutoksiin […].*
**Orpo** says *that he is ready for exceptional measures and even legistative changes […].* |

Table 1: The manually constructed rules for detecting quote-like syntactic structures.

| word | quote | entity |
|---|---|---|
| Kansainvälinen | B-INDIRECT+1 | B |
| rikostuomioistuin | I-INDIRECT+1 | I |
| aikoo | I-INDIRECT+1 | O |
| määrätä | I-INDIRECT+1 | O |
| Sudanin | I-INDIRECT+1 | B |
| presidentin | I-INDIRECT+1 | I |
| Omar | I-INDIRECT+1 | I |
| al-Bashirin | I-INDIRECT+1 | I |
| pidätettäväksi | I-INDIRECT+1 | O |
| , | O | O |
| kertoo | O | O |
| sanomalehti | O | B |
| New | O | I |
| York | O | I |
| Times | O | I |
| . | O | O |

Table 2: An example of sequence annotation for the BERT model.

|  | training | evaluation |
|---|---|---|
| articles | 1,172 | 287 |
| sentences | 22,949 | 5,097 |
| tokens | 252,006 | 59,076 |
| quotes | 3,854 | 984 |

Table 3: The sizes of datasets used in experiments.

| method | Pr | Re | F1 |
|---|---|---|---|
| rule-based | .85 | .78 | .82 |
| BERT | .92 | .90 | .91 |

Table 4: Results of quotation span detection without classification.

| method | indirect | | | direct | | |
|---|---|---|---|---|---|---|
| | Pr | Re | F1 | Pr | Re | F1 |
| rule-based | .75 | .66 | .70 | .93 | .86 | .89 |
| BERT | .84 | .84 | .84 | .96 | .94 | .95 |

Table 5: Results of quotation span detection and direct/indirect classification.

Thus, we present it as our best attempt given the current possibilities, but recognize the need for further work in this regard.

The annotated data assigns each quote to a single token representing the mention of the quote's source in the text. If the source is represented by a longer phrase, the syntactic head (wrt. dependency parsing) of this phrase should be selected according to the annotation guidelines. On the other hand, mentions of quote sources are typically entities annotated as parts of coreference chains, and thus the entire span is marked for the purpose of coreference annotation. Thus, by combining the quote and coreference annotations, we are able to obtain a span-to-span attribution relation for most cases. The exception are cases in which the quoted entity is mentioned only once in the article, and thus not annotated as a coreference chain.

Although the BERT model outputs sources as entity spans, the rule-based model points to a single token – the syntactic head, similarly to the gold standard annotation. In order to make the results comparable, we reduced the output of the BERT model to the first token of the span, and then evaluated a source annotation as correct if it either points to exactly the same token as the gold standard, or if it points to a token within the same coreference span. Thus, the model's ability to correctly identify the entire span is currently not evaluated, as it is not implemented in the rule-based method.

Table 6 presents results of the attribution evaluation in terms of the number of gold-standard quote tokens with **cor**rectly and **inc**orrectly recognized source, as well as **unrec**ognized source. The latter case occurs if either the token is not recognized as a quote at all, or it is recognized but without identifying the source. We report the accuracy as the ratio of correctly identified to all tokens.

The results indicate a small advantage of the rule-based model. In both cases, the main source of errors are the unrecognized annotations, rather than the incorrect ones. For the rule-based model this is typically due to quotes not being recognized at all (see low recall in Table 4), while for the BERT model there is a large amount of correctly identified quotes, for which the source could not be found. Of the 1990 recognized quotes, 646 (32%) are reported without source, compared to 13% (218/1633) for the rule-based model. The

| method | cor | inc | unrec | accuracy |
|---|---|---|---|---|
| rule-based | 7889 | 774 | 4996 | .58 |
| BERT | 7554 | 767 | 5338 | .55 |

Table 6: Results of attribution.

BERT model's ability to identify the source depends on the entity detection, for which the training data is incomplete (derived from coreference annotations only). Further, the model processes the text paragraph by paragraph and thus does not find a source mention that is outside of the paragraph containing the quote. These problems offer room for improvement in further work, and thus it can be expected that the BERT model will eventually outperform the rule-based one also in attribution.

## 6 Discussion and Further Work

Although we regard the work presented in the previous sections as a complete solution to a well-delimited problem, we see some potential for both incremental improvements, as well as work on further related tasks, that will be addressed in the future.

**Entity annotation and detection.** While designing our annotation project, we did not anticipate that a machine learning quote detection model will need to also detect entities that the quotes can be attributed to. We intended the coreference annotation to be used only in the further step (entity resolution). In result, entities that are mentioned only once were not annotated. The corpus could be improved by ensuring that at least tokens assigned as source to a quote are also annotated as an entity. This is expected to improve the BERT model's performance on entity detection, and thus quote attribution.

**Entity resolution.** While some works treat the problem of quote attribution to speaker mention in the text and entity resolution jointly (e.g. Muzny et al., 2017), in our opinion entity resolution is a complex task that is best treated separately. In addition to coreference resolution within one document, also matching the entities across documents could be considered there.

Coreference resolution can be done with BERT with state-of-the-art accuracy (Joshi et al., 2019). However, the setup is complicated as coreferences are typically long-range relations, so a sliding win-

dow approach needs to be used to mitigate BERT's limitation in text size. Furthermore, modeling relations with a neural model is not straightforward.

A related problem is that nested entities are possible and might be relevant, e.g.:

[[[Viron] metallityöväen liiton] puheenjohtaja Endel Soon]

[[[Estonia]'s metal workers' union]'s chairman Endel Soon]

In such case, coreferences and other quotes might also refer to the inner entities 'Estonia' or 'Estonia's metal workers' union'. For the present work, we disregarded nested entities as locally the outermost entity is typically the source of the quote it stands next to.

## 7 Conclusion

We have presented two methods for recognition of quotes in Finnish news media, along with an annotated corpus for training and evaluation. To our knowledge, our solution is the first one proposed for Finnish. We hope that the progress achieved on this task will facilitate more detailed large-scale quantitative analysis of voices in the Finnish news media.

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
