# OpenReview forum: "Detection and attribution of quotes in Finnish news media: BERT vs. rule-based approach"
_NoDaLiDa/2023/Conference — NoDaLiDa 2023_

### Official Review · Reviewer_jn1y · 2023-03-04
**A well-written and clear paper about detecting and attributing quotes in Finnish media texts.**

**Rating:** 8
**Confidence:** 4

**Review:**

This paper addresses the problem of finding quotes in Finnish news texts and attributing them to their source - the person, institution etc who has produced the quote.
This is the first attempt to do this kind of work for Finnish.

A gold-standard corpus containing ca 311 000 tokens in ca 28 000 sentences was annotated, two methods - rule-based and BERT-based - are used and their performance compared in the paper.

The results of quotation span detection are good (F1 is .82 for the rule-based method and .91 for BERT model). The numbers are lower for the task of attributing the quote to its source, the possible reasons and possible improvements are also discussed in the paper.

**Paper Type:**

Long paper

---

### Official Review · Reviewer_zgty · 2023-03-09
**Review on paper comparing BERT vs. a rule-based method for quotation detection and attribution**

**Rating:** 7
**Confidence:** 4

**Review:**

The authors compare two different approaches for detecting quotes in Finnish news text, and assigning them to the corresponding speaker. One of the methods corresponds to a rule-based approach operating on dependency trees, and the other corresponds to a machine learning approach based on fine-tuning a BERT language model with small amounts of annotated data. The models were evaluated on a new dataset (which the authors plan to release latter, and whose development followed a sound process that also considered assessing the inter-annotator agreement), and results showed that the BERT model can perform better, even if trained with few data. Overall, the paper is reasonably clear and well-written (although at some parts requiring additional clarifications), presenting an interesting study with a new dataset that the authors plan to release. Detailed comments are given next:

* In the description of related work, the authors state that "Scheible et al. (2016) have criticized the choice of CRFs for quote detection because of the Markov assumption they make." Ideally, this statement should be accompanied by a small description on why the Markov assumption is not appropriate for this task.

* In the proposed rule-based approach, the authors have used the "Turku Neural Parser" for dependency parsing, and the spacy system for analyzing the dependency trees. Why wasn't the same system used in both parts (e.g., spacy also features dependency parsing models for Finnish), given that this would simplify the software architecture? What is the performance/accuracy of the two different dependency parsers on Finnish text? This latter aspect would be interesting to include in the paper, given that errors in the dependency parsing will likely affect significantly the performance of the rule-based method.

* On what concerns the rule-based approach, the evaluation experiments could perhaps consider an ablated version of the method that features only the most simple method for detecting direct quotes (i.e., marking the sentences starting with a hyphen and the phrases enclosed in quotation marks). It would be nice to see how much the more complex rules, that use dependency parse trees, actually improve over this very simple baseline.

* The BERT model considers the tasks of quotation span identification and speaker assignment as a single classification problem (i.e., the same label space encodes BIO labels for tokens, and relative positions for the speakers). Perhaps the authors could have accessed a variant in which the two tasks are handled separately, e.g. as two different classification heads.

* Table 3, presenting a characterization for the dataset, could perhaps be complemented with information on the number of spans that correspond to speakers, and on the number of distinct speakers.

* The authors discuss a limitation with the use of the BERT model for performing quote attribution, related to the fact that "the annotated data assigns each quote to a single token representing the mention of the quote’s source in the text". I failed to understand why this constitutes a problem. Moreover, if the span annotations are based on word tokens, doesn't the use of BERT for the identification of quotation spans also involve a problem with tokenization (i.e., the ground-truth labels for word tokens need to be converted/matched to predicted labels in terms of BERT sub-word units).

* In connection to the BERT model, the authors also discuss the limitation that "the model processes the text paragraph by paragraph and thus does not find a source mention that is outside of the paragraph containing the quote." Isn't this problem even worse in the case of the rule-based method, in which the dependency parser analyses only individual sentences?

* In Section 6, the authors discuss a problem with the "entity annotation and detection," in which "entities that are mentioned only once were not annotated." I failed to understand why this was needed, and what is the relationship between this and the idea of using co-reference resolution only at a "further step."

* The paper has some minor problems in terms of presentation and correct English writing. The examples below correspond to sentences/phrases that can perhaps be slightly re-written.

It allows us to measure on a large scale, who is given
->
It allows us to measure, on a large scale, who is given

could be a useful element of such larger system as well.
->
could be a useful element of one such larger system as well.

latter case occurse if either
->
latter case occurs if either

**Paper Type:**

Long paper

---

### Official Review · Reviewer_Ysof · 2023-03-10
**Detection and attribution of quotes in Finnish news media: BERT vs. rule-based approach**

**Rating:** 8
**Confidence:** 3

**Review:**

In the paper "Detection and attribution of quotes in Finnish news media: BERT vs. rule-based approach" the authors introduce a new dataset for quote attribution in various Finnish news media and do a preliminary analysis of the performance on this new dataset with a classical rule-base system, as well as a BERT based system.
Quote attribution as an NLP task and datasets are rare, making this contribution all the more important.

The creation process for the dataset is well described, and their motivations and design decisions for their experiments are made very clear.
The evaluation for quote detection is straight forward and shows the superiority of the BERT-based approach; for the evaluation of the attribution of quotes  the authors themselves note some shortcomings in the annotation that led to some hacky evaluation decisions.
To counteract the authors take a closer look at the differences of mistakes made by the two systems to give a better idea for the actual performance, compared to the numbers presented

## Pros
- new dataset
- two "baseline" approaches
- overall clarity

## Cons
- attribution annotation could be better

## Minor fixes
017 wrt. -> the space after seems long, which should be fixed via "wrt.\ "
109 sec. -> Sec. and same spacing as above
126 sec. -> Sec.
230 wrt. spacing
232 The formulation is a bit misleading: could be understood as one annotator annotating 1460 articles while the remaining 9 annotated 40.
531 wrt. spacing


**Paper Type:**

Long paper

---

### Decision · Program_Chairs · 2023-03-17

Accept